# Etifoxine Restores Mitochondrial Oxidative Phosphorylation and Improves Cognitive Recovery Following Traumatic Brain Injury

**DOI:** 10.3390/ijms222312881

**Published:** 2021-11-28

**Authors:** Eilam Palzur, Doron Edelman, Reem Sakas, Jean Francois Soustiel

**Affiliations:** 1Eliachar Research Laboratory, Galilee Medical Center, Nahariya 2210001, Israel; eilam.palzur@gmail.com (E.P.); reem_sakas@hotmail.com (R.S.); 2Galilee Medical Center, Department of Neurosurgery, Nahariya 2210001, Israel; doronedelman@gmail.com; 3Azrieli Faculty of Medicine, University of Bar Ilan, Zafed 1311502, Israel

**Keywords:** mitochondria, traumatic brain injury, mitochondrial permeability transition pore, translocator protein, etifoxine

## Abstract

The opening of the mitochondrial permeability transition pore (mPTP) has emerged as a pivotal event following traumatic brain injury (TBI). Evidence showing the impact of the translocator protein (TSPO) over mPTP activity has prompted several studies exploring the effect of TSPO ligands, including etifoxine, on the outcome of traumatic brain injury (TBI). Mitochondrial respiration was assessed by respirometry in isolated rat brain mitochondria (RBM) by measurements of oxidative phosphorylation capacity (OXPHOS). The addition of calcium to RBM was used to induce mitochondrial injury and resulted in significant OXPHOS reduction that could be reversed by preincubation of RBM with etifoxine. Sensorimotor and cognitive functions were assessed following controlled cortical impact and compared in vehicle and etifoxine-treated animals. There was no difference between the vehicle and etifoxine groups for sensorimotor functions as assessed by rotarod. In contrast, etifoxine resulted in a significant improvement of cognitive functions expressed by faster recovery in Morris water maze testing. The present findings show a significant neuroprotective effect of etifoxine in TBI through restoration of oxidative phosphorylation capacity associated with improved behavioral and cognitive outcomes. Since etifoxine is a registered drug used in common clinical practice, implementation in a phase II study may represent a reasonable step forward.

## 1. Introduction

Traumatic brain injury (TBI) represents the leading cause of mortality and permanent disability in people under 45 years of age in western industrialized countries [1,2].

Following TBI, injured cells are threatened by a complex chain of interconnected events eventually leading to the death of potentially viable cells. The magnitude of this phenomenon, also known as secondary brain damage, has been stressed long ago by Reilly et al., who showed that the primary injury does not intimately correlate with final outcome, emphasizing the impact of the secondary brain injury on the fate of injured cells [3]. This observation, in turn, was the trigger for initiation of a vast and multidisciplinary research effort for a better understanding of the underlying mechanisms leading to secondary cell death and the development of novel therapeutic strategies. Encouraged by the promising results of basic research studies, clinical trials were initiated in increasing numbers up to the mid-1990s. However, the high expectations generated by encouraging laboratory data were not met by the deceiving results of subsequent clinical trials, making the pharmaceutical industry more reluctant to support expensive and adventurous research [4,5,6].

Among the deleterious events triggered by the injury, cerebral edema is considered as one of the most prominent threats during the clinical course of TBI [7,8]. Responsible for swelling of the brain encased within a rigid skull, edema results in elevation of the intracranial pressure (ICP), with subsequent impairment of cerebral perfusion and oxidative metabolism, presumably as a consequence of reduced oxygen delivery [9,10,11]. Accordingly, the relief of intracranial hypertension has remained the mainstay of the management of neurotrauma, either based on hyperventilation, hypertonic solutions, or decompressive craniectomy. However, although clinical evidence does support the beneficial effect of ICP control on cerebral perfusion [12,13,14], this effect does not correlate in most instances with concurrent improvement of oxidative metabolism, suggestive of its non-ischemic origin [15,16].

Although cerebral edema may represent the consequence of disruption of the blood-brain barrier known as vasogenic edema, cellular swelling or cytotoxic edema is considered to be prominent during the early post-traumatic period [17]. Cytotoxic edema results from a massive cellular influx of Na^+^, Ca^+^, and water triggered by the glutamate-induced post-synaptic activation of *N*-Methyl-*D*-Aspartate, AMPA, and kainic acid receptors. As a physiological protective mechanism, calcium is absorbed within the mitochondrial matrix, where its accumulation eventually results in protein denaturation, leading to electron transport chain dysfunction and subsequent loss of the proton gradient across the mitochondrial membrane. Dissipation of the mitochondrial transmembrane potential results, in turn, in permeabilization of the internal mitochondrial membrane through the opening of the mitochondrial permeability transition pore (mPTP), a phenomenon known as mitochondrial permeability transition that causes an unrestricted influx of ions and fluid within the mitochondria with subsequent and irreversible structural damage to the mitochondrial membrane. Occurring in parallel as the energy crisis develops, ATP depletion is be responsible for the failure of energy-dependent ionic pumps, leaving the influx of water and solutes uncontrolled [18,19]. Cytotoxic edema and failure of the oxidative metabolism may thus represent the two different but interconnected aspects of the same mitochondrial damage. As such, it may be hypothesized that therapeutic mitochondrial-protective measures may restore at least a subnormal energy state of the injured cells and thus contribute to the control of cytotoxic edema by restoring the cellular homeostasis capability, leading to improved neurological outcomes without mechanically interfering with the intracranial pressure.

Although the exact structure of the mPTP remains a matter of controversy, several molecules have been advocated as possible regulators of the pore activity. Among these, the 18 kDa translocator protein (TSPO) has initially gained increasing attention by its location at the outer mitochondrial membrane and its co-immunoprecipitation with the adenine nucleotide translocator and the voltage-dependent anion channel previously believed to be core constituents of the mPTP [20,21]. The hypothesis of a possible regulatory role of the TSPO over the process of mitochondrial permeability transition has been supported by experimental studies showing that the addition of Ro5-4864, a TSPO ligand, to mitochondrial pellets resulted in a protective effect against known mitochondrial noxious agents such as calcium and Bax, with preserved mitochondrial membrane polarization, and decreased activation of caspase 9 [22,23]. These findings were further supported by experimental studies showing that treatment with Ro5-4864 was associated with enhanced neuronal survival and improved oxidative metabolism expressed by a lower cerebral lactate/pyruvate ratio [23,24]. However, since the clinical prospect of Ro5-4864 in TBI remains limited because of its epileptogenic properties [25,26], attention was drawn to alternative potentially beneficial TSPO ligands such as etifoxine. Etifoxine is currently used in standard clinical practice to manage adjustment disorder with anxiety [27] though it was also shown to increase concentrations of pregnenolone, progesterone, 5alpha-dihydroprogesterone, and allopregnanolone in the plasma and the brain of etifoxine-treated animals, indicating both its activity at the TSPO level and its ability to cross the blood-brain barrier [28]. Furthermore, etifoxine has been shown to reduce cerebral edema in different models of brain injury [29,30,31], and more recently to improve both pathological and neurological outcomes in etifoxine-treated animals submitted to controlled cortical impact models of TBI [32].

Accordingly, the purpose of the present study was to further investigate the potentially beneficial effect of etifoxine at the mitochondrial level and to explore the possible correlation of such a mitochondrial protective effect with the sensorimotor and cognitive functional outcome of animals exposed to TBI.

## 2. Results

### 2.1. Mitochondrial Respiration

No significant difference could be found between the oxygen consumption rates in the different groups following the addition of succinate (Figure 1 and Figure 2A). Oppositely, the addition of calcium was associated with a significant decrease of the rate of oxygen consumption by isolated mitochondria in the presence of ADP (*p =* 0.0032, control vs. calcium group Tukey-Kramer *p <* 0.05) that was partially reversed by incubation of mitochondria with etifoxine (Figure 1 and Figure 2B), although the difference between the calcium and the etifoxine group did not reach statistical significance (calcium vs. etifoxine group Tukey-Kramer *p* > 0.05). Similar observations could be made when analyzing the OXPHOS capacity defined by the rate of oxygen consumption after the addition of pyruvate in the presence of ADP saturation (*p =* 0.013, Figure 1 and Figure 2C). A substantial and significant decrease (24.3%) in OXPHOS capacity could be observed in comparison with intact mitochondria (Figure 1 and Figure 2C, control vs. calcium group Tukey-Kramer *p <* 0.05), despite the relatively short time of exposure of mitochondria to calcium until the addition of pyruvate (~10 min). Remarkably, this deleterious effect of calcium could be almost entirely reversed by etifoxine, restoring an OXPHOS capacity close to that of the control group and significantly different from that of the calcium group (etifoxine vs. calcium group Tukey-Kramer *p <* 0.05). Finally, although analysis of the differences between the oxygen consumption rates measured in the different groups following the addition of rotenone showed some statistical trends (*p =* 0.047) similar in nature with that of ADP and pyruvate, the differences observed between the groups failed to reach statistical significance (Figure 1 and Figure 2D). As anticipated, no differences could be found between groups following the addition of antimycin A (Figure 1 and Figure 2E).

### 2.2. Mitochondrial Swelling

As anticipated, the addition of calcium to non-injured mitochondria resulted in the loss of mitochondrial membrane permeability expressed by a substantial decrease in light scattering at 540 nm consistent with increased mitochondrial volume that correlated with Ca^2+^ concentration (*p* < 0.001 Repeated Measures ANOVA Tukey-Kramer main effect of Ca^2+^ concentration, Figure 3). In contrast, pre-incubation of isolated mitochondria with 80 µM of etifoxine significantly delayed mitochondrial swelling with significantly slower rate of decrease in light-scattering (*p* < 0.001 Repeated Measures ANOVA main effect of group and combined effect of group and concentration, Figure 3).

### 2.3. Cognitive Outcome

As expected, analysis of water maze testing performance showed a similar trend of decrease of latencies to the platform in both non-treated and treated animals during the training period prior to injury (Figure 4A).

Following TBI, there was no significant difference in performance or latency to the platform between the groups on the first day (Figure 4B), though thereafter the pace of cognitive improvement was significantly higher in the etifoxine-treated group (Repeated-Measures ANOVA, combined effect of day and group, *p =* 0.0098).

In order to examine the combined effect of group and time-points interaction in cognitive performance expressed by latency to the platform, repeated measures ANOVA analyses were used to compare performance in the five time-points (within-subject comparisons) in each group. There was a decrease in latency to platform in the two groups (*p <* 0.001 for both groups, main effect of time) with contrast analyses showing a significant decrease in latency to platform between day one and two and day two and three in the treatment group only (*p <* 0.001). However, in the control group, there was no significant difference in performance between the first three days, but only a significant difference between the first day and the last day (*p <* 0.05). In addition, Tukey-Kramer post-hoc analysis showed a significant difference in latency to platform between the non-treated and treated groups at day two post-injury (*p <* 0.05; Figure 4B).

### 2.4. Motor Outcome

Statistical analysis of the performance in the rotarod test in an eight time-points follow-up period revealed a significant and equal decrease in motor performance in the first two days post-injury as expressed by the latency to fall in the two groups of animals (Figure 5, Repeated-Measures ANOVA, main effect of time, *p <* 0.0001). Although data analysis was suggestive of some improvement in etifoxine-treated animals in comparison with performance of rats in the control group, the difference did not prove to be statistically significant between the two groups (*p* > 0.05).

## 3. Discussion

The search for a potential therapy for the management of traumatic brain injury has been ongoing for decades. Promising experimental results, however, failed to successfully translate to human TBI trials and pushed the scientific community back to ICP-targeted management. This pragmatic and very cautious approach in the development of novel therapeutic strategies has been even further and recently emphasized by the publication of the findings of Operation Brain Trauma Therapy, led by a group of distinguished scientists long dedicated to the challenge of TBI, leaving a very narrow window for initiation of new human clinical trials [33].

Increasing expectations from ICP-targeted management eventually prompted two successive large international multicenter studies, RESCUEicp and DECRA [34,35], that did confirm that relief of uncontrolled intracranial hypertension was an effective measure of preventing death due to cerebral herniation although the hope for improved functional outcome was not met by the findings of both studies. In a cohort of 36 severe TBI patients who underwent decompressive craniectomy for uncontrolled ICP elevation, we were able to show a beneficial impact of ICP relief on the odds of survival and to provide evidence that brain decompression was associated with improvement of cerebral blood flow (CBF), though oxidative metabolism was left unaffected by surgery [14]. This observation was consistent with that of a bicenter randomized controlled study on the effect of hypertonic solutions that showed that ICP relief was associated with significant CBF enhancement, but not with any improvement of oxidative metabolism [13], supporting the non-ischemic origin of post-traumatic energy crisis reported by several studies [12,15,16].

Prevention or reduction of the magnitude of the energy crisis triggered by the injury through regulation of the activity of the mPTP has drawn increasing attention during the past decade and emerged as an alternative route of neuroprotection [36,37]. Among the most documented and potent inhibitors of the mPTP opening, cyclosporin A proved to significantly reduce the extent of brain damage in experimental models of brain injury, decrease mitochondrial dysfunction and improved both sensorimotor and cognitive outcome caused by TBI [38,39,40,41]. In their pioneer work, Okonkwo and Povlishock showed that cylosporin A significantly reduced mitochondrial swelling and axonal damage, hypothesizing that preserved energy production could support activation of ionic membrane pumps and restore cellular homeostasis [39]. In a recent experimental study, we were able to provide evidence that this mitochondrial protective effect of cyclosporin A was indeed associated with significantly reduced brain water content and neuronal swelling and, as a consequence, decreased ICP significantly [42]. In the same study, the hypothesis of a causative relationship between mitochondrial dysfunction and cytotoxic edema could be further demonstrated by the opposite and deleterious effect of the ATP synthase inhibitor oligomycin B on the same outcome measures in the same conditions. However, the enthusiasm for implementation of cyclosporine A in the management of human TBI has been recently reduced by the findings reported by Dixon et al. as part of Operation Brain Trauma Therapy. These findings were characterized by absence of any beneficial cognitive effect in the three different models of TBI used in the study (parasagittal fluid percussion injury, controlled cortical impact, and penetrating ballistic-like brain injury), relatively high signs of toxicity, and a narrow therapeutic index [43].

Within the spectrum of mPTP modulators, the TSPO has drawn increasing attention. Long known as a sensitive marker of traumatic brain damage [44,45], the TSPO has emerged as a highly probable regulator of the mitochondrial permeability transition, capable of promoting cell death in a variety of malignant cell cultures exposed to PK11195 while treatment with Ro5-4864, a distinct TSPO ligand, resulted in enhanced cell survival in different models of brain injury [23,24,46]. More specifically, Ro5-4864 was shown to protect mitochondria both structurally and functionally, enhanced neuronal survival and axonal integrity and reduced brain edema. Despite these encouraging results, the clinical prospect of any translational study involving Ro5-4864 remains limited by the epileptogenic side effects of the drug, hence the need of an alternative and less toxic TSPO neuroprotective ligand [25,26]. Indeed, benzodiazepines do not represent the entire spectrum of TSPO ligands, and several drugs have been developed, including benzothiazepines, phenoxyphenyl acetamides, isoquinoline carboxamides, indol acetamides, imidazopyridine acetamides, pyrazolo-pyrimidine acetamides, and indol-3-ylglyoxylamides. Among TSPO ligands, etifoxine represents the progenitor of the benzoxazines that proved to bind to TSPO with a high affinity and to induce a significant increase in the concentrations of pregnenolone, progesterone, 5alpha-dihydroprogesterone, and allopregnanolone in the plasma and in the brain of treated animals [28]. The renowned neuroprotective effect of steroids generally and that of progesterone specifically prompted several experimental studies showing a protective effect in different animal models of neurological disorders including multiple sclerosis [46], stroke [30,31] and peripheral nerve diseases [47,48].

Our findings demonstrate that the deleterious effect of calcium on mitochondrial oxidative phosphorylation capacity could be substantially reversed by incubation of mitochondrial pellets with etifoxine 30 minutes prior to calcium addition. Exposure of isolated rat brain mitochondria (RBM) to increasing concentrations of calcium recreates the pathophysiological conditions induced by TBI and characterized by calcium accumulation within the mitochondrial matrix with subsequent dissipation of the mitochondrial transmembrane potential and is therefore considered as an accepted model of mPTP opening [22]. In addition, incubation of RBM with etifoxine was associated with a significant reduction in calcium-induced mitochondrial swelling, further supporting the hypothesis of a protective effect of etifoxine on mitochondrial membrane permeability [49], and in accordance with the mitochondrial protective effect shown by Ro5-4864 in our previous studies. An important limitation of the present study, however, is represented by the absence of a definite assessment of calcium retention capacity with direct measurements of PT-induced calcium release in the suspension buffer so that no firm conclusion regarding the status of the mPTP should be drawn. Furthermore, the possible regulating role of TSPO on the mPTP has been recently challenged by the absence of any morphological or functional difference observed in TSPO-null liver mitochondria [50]. Importantly, the effect of TSPO ligands on mPTP induction proved to be strictly identical in native and TSPO-null mitochondria in the same study, suggestive of a distinct and indirect mechanism of action of TSPO ligands on mPTP activity. Indeed, previous experimental studies have shown that ligands characterized by high TSPO affinity also bind and inhibit F(1)F(0)-ATPase activity, suggesting that some of the mitochondrial responses observed with these ligands commonly attributed to TSPO modulation may in fact represent the result of mitochondrial F(1)F(0)-ATPase inhibition [51,52]. Such a hypothesis may, in turn, account for the impact of etifoxine on oxidative phosphorylation presently reported without directly interfering with the mPTP.

Regardless of the exact mechanism of improved oxidative metabolism mediated by etifoxine and following the guidance of the Operation Brain Trauma Therapy [53], a more detailed attention was brought to functional outcomes in the present study with the implementation of sensorimotor and cognitive assessment tools. Although there was no evidence of any positive effect on the motor performance as explored by the rotarod, treatment with etifoxine was associated with a clear and significant improvement of the learning curve, expressed by the faster decrease of the latency to platform with the Morris water maze testing. This observation is in accordance with the improvement in modified neurological severity score shown in etifoxine-treated animals in our previous study, as well as with a similar observation reported by Li et al. in etifoxine-treated animals following intracerebral hemorrhage [31].

## 4. Methods

### 4.1. Animals

All described experiments were performed according to the “Institutional Animal Ethical Committee”—US National Research Council (8th edition, 2011) and were approved by the Committee for Animal Research Inspection of Bar-Ilan University (approval #27-05-2016). During the study, the animals were housed in groups of two to three rats in a sterilized solid bottom cage with contact bedding under controlled temperature and 12:12 h light/dark cycle and maintained on a standard pellet diet water supplied ad libitum. All efforts were made to keep animals suffering to a minimum and to lower the number of animals used as much as possible. Accordingly, sham studies were deliberately discarded from the study design based on our own experience as well as that of the literature showing that sham experiments are constantly and mostly innocuous and do not therefore justify the additional sacrifice of a substantial number of rats.

### 4.2. Brain Injury Model

#### 4.2.1. Model Description

The brain injury model was based on a modified controlled cortical impact (CCI) injury described by Dean et al. [54]. Briefly, male Sprague-Dawley rats (300–350 g) were anesthetized using 2–4% isoflurane in 100% oxygen within an induction chamber. Following the induction of deep anesthesia upon confirming the lack of pain responses, animals were transferred and fixed in a stereotaxic rat frame (Stoelting, Wood Dale, IL, USA) while maintained under anesthesia through a nose cone with 2–4% isoflurane using the SomnoSuite anesthesia delivery system (Kent Scientific, Torrington, CT, USA). Animals’ body temperature was maintained at 37 °C with the use of an isothermal pad. Next, a longitudinal incision was made down the midline of the head to expose the skull. Following skull exposure, a 6 mm diameter craniotomy was made on the right hemisphere, 0.5 mm lateral to the sagittal suture, and midway between the lambda and bregma sutures. Care was taken not to disrupt the dura. Animals with damaged dura were excluded. The exposed dura was then impacted with the Impact One Stereotaxic Impactor for CCI (Leica Biosystems, Wetziar, Germany), using a 5 mm diameter tip at a velocity of 5 m/s and a dwell time 100 msec at a depth of 1mm. During the impact, anesthesia was turned off briefly for a few seconds allowing a breath of pure oxygen before the injury as a preventive measure against possible post-traumatic apnea. The scalp wound was then sutured, and the rat allowed to recover from anesthesia in an individual cage.

#### 4.2.2. Animal Grouping and Treatment

Following the injury, animals were allocated into two groups of seven rats each as follows: group 1 (control/vehicle group)—treated with Tween^®^ 80 (Sigma Aldrich, St Louis, MO, USA) 1% in saline; group 2 (treatment group)—treated etifoxine (Biocodex, Gentilly, France) dissolved in Tween^®^ 80 at a dosage of 50 mg/kg. Animals of both groups received an intraperitoneal injection (vehicle in the control group and etifoxine in the treatment group) at 4, 12, 24 and 48 h post injury. The dosage used in this study was based on our previous study showing a maximal beneficial effect obtained with 50 mg/kg [32].

### 4.3. Isolation of Mitochondria from Rat Brain

Rat brain mitochondria were isolated following the protocol described by Sumbalova et al. [55]. Briefly, tissue samples were harvested from the parietal region of the cerebral hemisphere of uninjured animals. Tissue samples were disposed of and washed in phosphate buffer solution for five minutes at 4 °C, and then 100–180 mg of brain tissue was cut into small pieces and homogenized in 10 folds (1–1.8 mL) of ice-cold isolation medium containing (0.32 M sucrose, 1 mM K^+^-EDTA, 10 mM TRIS–Cl, pH 7.4) with the addition of 2.5 mg/mL of BSA, using a glass/teflon homogenizer. Brain homogenate was transferred to a 2ml Eppendorf tube and centrifuged at 4 °C for 10 min at 1000× *g*. The supernatant was centrifuged for 10 min at 6200× *g* and 4 °C. The mitochondrial pellet was washed twice with the isolation medium without BSA and resuspended in a small volume of the same medium. The concentration of mitochondrial proteins was determined according to Bradford assay using the Pierce, Coomassie Plus Protein Assay with BSA as a standard. Finally, concentrations of 0.25 mg/mL of mitochondria protein were used for respirometric measurements.

### 4.4. Mitochondrial Respiration Measurements

Mitochondrial respiration was measured with high-resolution respirometry OROBOROS Oxygraph-2k (Oroboros Instruments, Innsbruck, Austria) at 37 °C in 2 mL Mitochondrial respiration medium containing 110 mM sucrose, 60 mM K^+^-lactobionate, 0.5 mM EGTA, 3 mM MgCl_2_, 20 mM taurine, 10 mM KH_2_ PO_4_, 20 mM HEPES adjusted to pH 7.1 with KOH at 37 °C, and 1 g/L BSA essentially fatty acid-free.

During the respirometry assay, the oxygen concentration (μM), as well as oxygen flux per tissue mass (pmol O2/s/mg), was measured and recorded. Following the addition of 0.25 mg/mL mitochondrial protein, the respiration essay started with the complex-II-linked substrate succinate (10 mM) to induce first resting respiration, representing a proton leak-driven respiration (LEAK). Then the respiration buffer was supplemented with 2mM ADP and 5mM pyruvate, simulating the activated state of oxidative phosphorylation (OXPHOS) at saturating concentrations of ADP. Inhibition of complex I by rotenone (0.5 μM) thereafter allowed for measurement of complex-II-linked electron transfer capacity. Finally, to control whether other oxygen-consuming processes are involved in mitochondrial respiration, complex III was inhibited by antimycin A (2.5 μM) [56].

According to the manufacturer’s recommendations, the oxidation fluxes were compared after correction for residual oxygen consumption (ROX), corrected automatically by DatLab^®^ software for instrumental background [56].

In order to simulate the mitochondrial damage that occurs in cytotoxic edema due to calcium overload, calcium (6 mM) was added to the mitochondrial respiration medium in the control group before initiation of the successive measurements described above. Mitochondrial pellets constituting the treated group were incubated for 30 min with etifoxine (80 µM) before addition of calcium [32]. The rates of oxygen consumption in the LEAK and OXPHOS states were then compared between the two groups of seven mitochondrial samples each.

Figure 6 depicts a typical recording of oxygen consumption trends in the control group throughout the successive substrates’ additions.

### 4.5. Assessment of Ca^2+^-Induced Mitochondrial Swelling

The activation of the mitochondrial permeability transition pore was determined by Ca^2+^-induced swelling of isolated brain mitochondria according to the technique thoroughly described by Quinlan et al. [49]. According to this technique, mPTP induction mediated by calcium overload results in mitochondrial swelling that can be quantified spectrophotometrically as a decrease in the absorbance at 540 nm. For this purpose, mitochondria from non-injured brains were isolated as described above and resuspended in a swelling buffer, which contained 10 mM HEPES, pH 7.4, containing 250 mM sucrose, 1 mM ATP, 0.08 mM ADP, 2.5 mM sodium succinate, 1 μM rotenone, 2 mM K_2_HPO_4_, and 1 mM DTT to a final protein concentration of 500 μg/mL. A pore opening was then induced by addition of Ca^2+^ in the form of CaCl^2^. Mitochondrial pellets of each sample were tested twice at increasing concentration of CaCl^2^ (50, 150 and 300 μM, one at a time) while absorbance was measured and averaged. Trends of light scattering decrease were continuously recorded until stabilization for each Ca^2+^ concentration with a spectrophotometer. The same experiment was repeated after previous incubation of isolated mitochondria in the assay buffer following the addition of 80 µM etifoxine for 10 min prior exposure to calcium. This entire set of measurements was repeated with five distinct mitochondrial samples obtained for distinct animals. Absorbance trends obtained in the five different sets of two averaged measurements performed at each calcium concentration, with and without etifoxine, were compared with a Repeated Measures ANOVA.

### 4.6. Cognitive Outcome

The Morris water maze was used to measure a new spatial reference memory task 1–five days before and one to five days after brain injury. Briefly, animals were placed into a pool containing a hidden platform, while the time for the rat to reach the platform (escape latency) was recorded.

The maze is a large circular pool 1.5 m in diameter, 40 cm deep, filled with 26 ± 2 °C water. A clear plexiglass platform placed at 30.5 cm height from the pool bottom was submerged in the middle of one quadrant of the pool. Swim path and latency to find the hidden escape platform was monitored using a computer-controlled tracking system (EthoVisionXT, version 15, Noldus Information Technology, Inc., Leesburg, VA, USA). Spatial cues were placed within the test room. Each day, animals completed four trials, with each trial starting, in random order, from one of the four-quadrants (north, south, east, and west) of the pool. Animals were placed into the pool in a randomized order. Rats not finding the hidden platform within 120 s were placed directly on the platform for 30 s. Rats were placed in a warmed cage during the 4-min inter-trial interval. The latency to the platform, average velocity, and time spent in the target quadrant with the platform was recorded. Animals were tested for five consecutive days before TBI as a baseline measure, while the same protocol with the platform placed at a different quadrant was used for five days after TBI for outcome assessment.

### 4.7. Motor Outcome

Motor coordination and balance were assessed using the Rotarod (Rat Rotarod NG, Model 47750; Ugo Basile, Varese, Italy). The apparatus consists of a rotating rod (6 cm diameter) with machined grips, divided into four equal 8.7 cm wide sections raised 30 cm above trip boxes. Animals were trained with the Rotarod one week before injury. In training trials, rats were placed on the rod, which rotated at a constant speed of 10 rpm. The training trial continued until the rat could stay on the rod for 60 consecutive seconds without falling, turning around, or clinging to the rod. If they fell from the rod or turned around, they were placed back on the rod correctly, and the timer restarted. In test trials, an accelerating protocol was used, where the speed of rotation increased from 10 to 40 rpm for 300 s. Each trial was terminated if an animal fell, clung, rotated for two complete rotations, or remained on for more than 300 s. Latency to falls were automatically recorded for each trial. The average of the three trials was calculated and used for analysis. Baseline values were recorded 24 h before CCI. The Rotarod apparatus was wiped with 30% ethanol and allowed to dry completely between subjects. Animals were tested for nine days post-TBI.

### 4.8. Data Analysis

Variations in the different outcome measurements in the different groups were explored by One Way or Repeated Measures models of ANOVA according to data requirements. Whenever appropriate, post hoc analysis of differences noted between groups were tested using the Tukey–Kramer multiple comparison procedure. A *p*-value of less than 0.05 was considered significant.

## 5. Conclusions

The findings of the present study further confirm the results of our previous study that showed enhanced neuronal survival and reduced tissue loss in a dose-dependent fashion in etifoxine-treated animals. Our results are indicative of a protective effect of etifoxine over TBI-induced mitochondrial damage, even though the exact mechanism of mitochondrial protection remains a matter of controversy. Nevertheless, restoration of oxidative phosphorylation capacity was associated as hypothesized with improved behavioral and cognitive outcomes. Since etifoxine is a registered drug used in common clinical practice, implementation in a phase II study may represent a reasonable step forward.

## Figures and Tables

**Figure 1 ijms-22-12881-f001:**
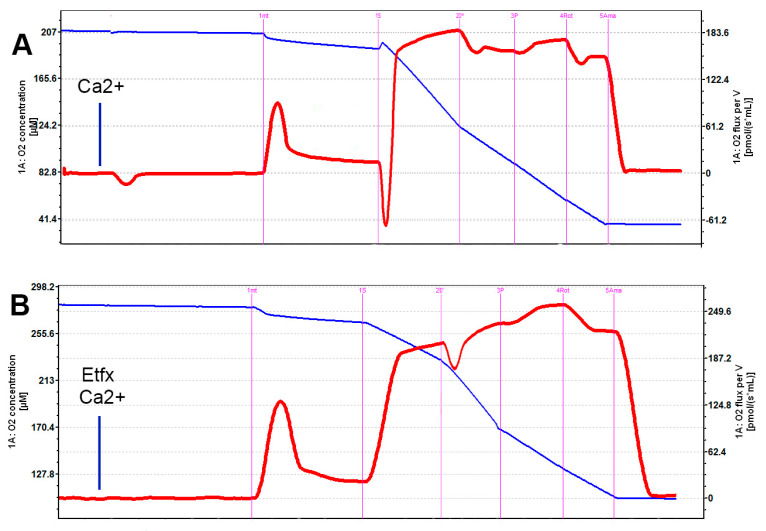
Typical oxygraph recordings of oxygen consumption rate by intact RBM throughout the successive phase of the testing following addition of 6 mM Ca^2+^ (**A**) and 6 mM calcium and 80 µM etifoxine (**B**). Following stabilization of the medium, mitochondria are added (1mt), then succinate (1S), ADP (2D), pyruvate (3P), rotenone (4Rot) and actinomycin A (5Ama). Oxygen consumption rate is represented by the red line while oxygen concentration within the oxygraphy chamber is shown by the blue line. For each level, data is extracted by DataLab^®^ v7.4 software. Note that the time scale has been adapted for the graphs to grossly match.

**Figure 2 ijms-22-12881-f002:**
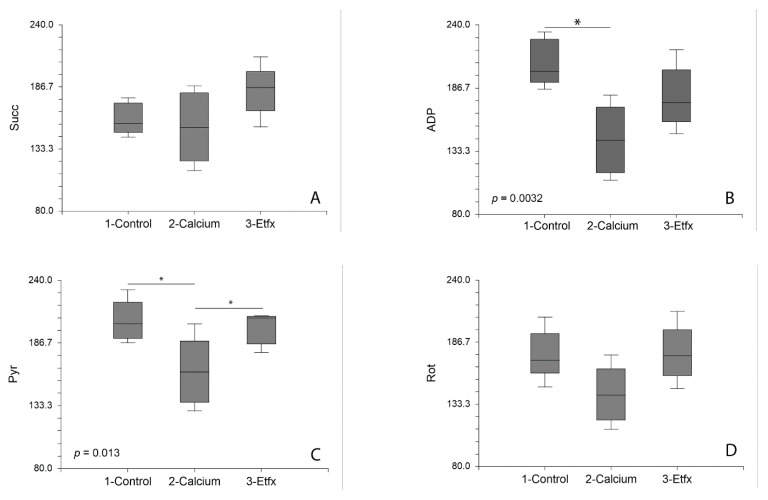
Exposure of RBM to calcium resulted in impairment of the oxidative phosphorylation expressed by significantly reduced oxygen consumption rates following addition of ADP and pyruvate (**B**,**C**) in contrast with the absence of significant changes across the three groups (control, calcium, calcium and etifoxine) following succinate (**A**), rotenone (**D**) and antimycin A (**E**). Pre-incubation with etifoxine, however, could partially (ADP) or completely reverse this effect of calcium on oxidative phosphorylation. *p*: *p*-value indicates the main effect of group in One-way ANOVA. *: Results of post-hoc analysis between groups with Tukey-Kramer’s test when appropriate.

**Figure 3 ijms-22-12881-f003:**
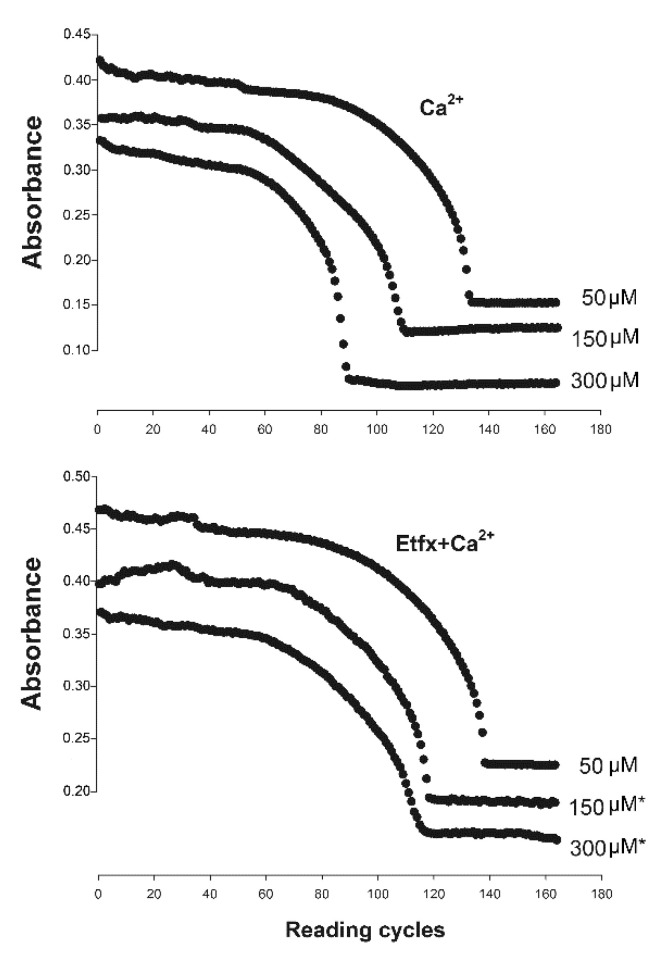
Respective trends of changes in light-scattering in intact mitochondria at 540 nm following addition of Ca^2+^ at increasing concentration of 50, 150 and 300 µM without (**upper**) and with (**lower**) pre-incubation with etifoxine (etfx). Addition of etifoxine to the suspension buffer 10 min before calcium addition resulted in enhanced resistance to calcium-induced damage with increased delay in mitochondrial swelling expressed with decreasing levels of light-scattering. The difference proved to be significant at 150 and 300 µM (Repeated Measures ANOVA, *p <* 0.001 main effect of group and concentration). *: Tukey-Kramer Multiple-Comparison Test etifoxine vs. control at 150 and 300 µM, *p <* 0.05.

**Figure 4 ijms-22-12881-f004:**
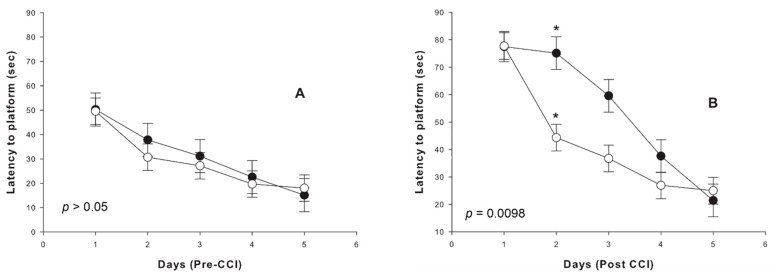
Morris water maze testing showed similar learning curves in the two groups prior to injury (**A**). Following CCI, however, treatment with etifoxine was associated with a clear and significant improvement of the learning curve expressed by faster decrease of the latency to platform (**B**). Vehicle-treated group: black dots (±standard error). Etifoxine-treated group: white dots (±standard error). *p*: main effect of group in One-way ANOVA. *: Results of post-hoc analysis between groups with Tukey-Kramer’s test when appropriate.

**Figure 5 ijms-22-12881-f005:**
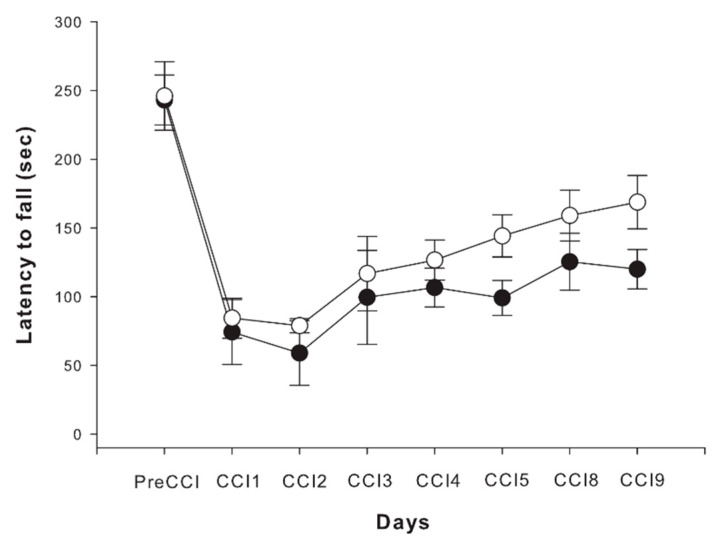
Comparative trends of sensorimotor recovery following CCI in vehicle-treated (black dots ± standard error) and etifoxine-treated animals (white dots ± standard error) tested by rotarod test and expressed by latency to fall (sec). Although data analysis was suggestive of some improvement in etifoxine-treated animals, the difference did not prove to be statistically significant between the two groups (*p* > 0.05).

**Figure 6 ijms-22-12881-f006:**
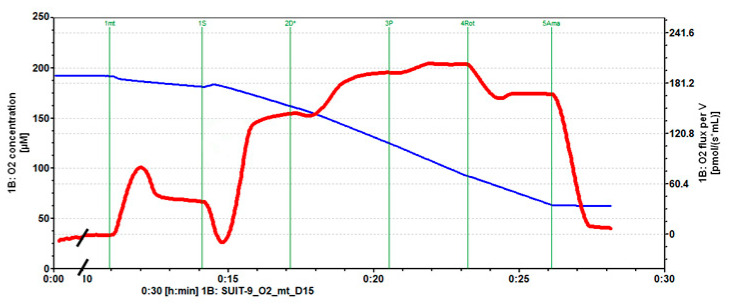
Typical oxygraph recording of oxygen consumption rate by intact RBM throughout the successive phase of the testing. Following stabilization of the medium, mitochondria are added (1mt), then succinate (1S), ADP (2D), pyruvate (3P), rotenone (4Rot) and actinomycin A (5Ama). Oxygen consumption rate is represented by the red line while oxygen concentration within the oxygraphy chamber is shown by the blue line. For each level, data is extracted by DataLab^®^ software. Note that part of the medium stabilization phase has been cropped (black slashes).

## Data Availability

The data presented in this study are openly available in FigShare under the name of the corresponding author.

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
