# Peer review of "Etifoxine Restores Mitochondrial Oxidative Phosphorylation and Improves Cognitive Recovery Following Traumatic Brain Injury"

_ijms, 2021, doi:10.3390/ijms222312881_

Round 1

Reviewer 1 Report

The authors report on the beneficial role of etifoxine on mitochondrial oxidative phosphorylation and cognitive recovery following TBI in the rat.  This well-designed and executed study reports valuable information on a new application for an already approved drug, etifoxine.  However, there are some issues that need clarification.

It is evident that the authors based their work on the anxiolytic properties of etifoxine acting via the mitochondrial TSPO. Typically the Morris Water Maze is a spatial memory behavioral test for rodents, but it is based on the idea that being forced to swim stresses the animals out, and therefore, they will try to find the submerged platform as soon as possible to elevate themselves out of the water. Stress has been shown to impair spatial memory retrieval, and the use of an anxiolytic could decrease the stress in rodents. Because of this, I'm wondering if etifoxine could be influencing the behavioral results due to its effect on GABA instead of the effects on TSPO or a combination of both.  The authors should discuss this point.

Do the authors know whether etifoxine treatment would restore oxidative phosphorylation and mitochondrial function if administered post calcium treatment? The data presented in the manuscript report only on mitochondria pre-treated with etifoxine.

Several publications have shown that TSPO ligands and TSPO deficiency can inhibit microglial activation and thus influence neuroinflammation. Do the authors know whether the observed effects were seen across all cell types or specific to neurons or microglia.

Author Response

Reviewer #1

General comment: "The authors report on the beneficial role of etifoxine on mitochondrial oxidative phosphorylation and cognitive recovery following TBI in the rat.  This well-designed and executed study reports valuable information on a new application for an already approved drug, etifoxine.  However, there are some issues that need clarification."

 Answer: The authors wish to thank this reviewer for the attention brought to an important aspect of this study, namely the potential impact of an anxiolytic drug on stress issues in cognitive assessment.

Comment: "It is evident that the authors based their work on the anxiolytic properties of etifoxine acting via the mitochondrial TSPO. Typically the Morris Water Maze is a spatial memory behavioral test for rodents, but it is based on the idea that being forced to swim stresses the animals out, and therefore, they will try to find the submerged platform as soon as possible to elevate themselves out of the water. Stress has been shown to impair spatial memory retrieval, and the use of an anxiolytic could decrease the stress in rodents. Because of this, I'm wondering if etifoxine could be influencing the behavioral results due to its effect on GABA instead of the effects on TSPO or a combination of both.  The authors should discuss this point."

Answer: As postulated by reviewer # 1, we used the Morris water maze test to study the effect of etifoxine on spatial memory and learning following TBI. However, we were aware that this test uses aversive stimuli rather than reward, which would make it stressful for the animal, as was suggested by Harrison et al.,1. Although swimming is less stressful for rats than mice 2, we were concerned about this fact; therefore, we started the post-TBI trial after completing the treatment with etifoxine. Zheng et al.,3 showed that stress improved rats’ performance in the Morris water maze test and documented the role of the GABAA receptor in this process. In general, GABAA receptor agonists impair memory 4, 5. Thus we thought that the anxiolytic property of etifoxine as well as the fact that it acts as a GABAA receptor agonist 6, 7 would reduce the anxiety hance the motivation to escape to the platform together with impairment in memory performance by resulting in an increase in latency time in the treated group compared to the untreated group. Since our results were the opposite, we assumed that etifoxine improved the cognitive task via the TSPO rather than GABA. Additionally, the neuroprotective effect of etifoxine by acting on TSPO is supported by our previous work that demonstrated its salvage property on neurons, implicated by the reduction in Lesion Volume and the histology findings 8.

  1. Harrison FE, Hosseini AH, & McDonald MP (2009). Endogenous anxiety and stress responses in water maze and Barnes maze spatial memory tasks. Behavioural brain research, 198 (1), 247-51.
  2. Whishaw, I.Q. (1995). "A comparison of rats and mice in a swimming pool place task and matching to place task: some surprising differences." Physiology & Behavior. 58 (4): 687–693.
  3. Gang Zheng, Xueping Zhang, Yaoming Chen, Yun Zhang, Wenjing Luo, Jingyuan Chen (2007). Evidence for a role of GABAA receptor in the acute restraint stress-induced enhancement of spatial memory. Brain Research, 1181, 61-73.
  4. Zarrindast MR, Bakhsha A, Rostami P, Shafaghi B. (2002). Effects of intrahippocampal injection of GABAergic drugs on memory retention of passive avoidance learning in rats. Journal of Psychopharmacology. 16(4):313-319M.R
  5. Zarrindast, T Shamsi, P Azarmina, P Rostami, B Shafaghi (2004). GABAergic system and imipramine-induced impairment of memory retention in rats. European Neuropsychopharmacology, 14 (1), 59-64.
  6. Mattei C, Taly A, Soualah Z, Saulais O, Henrion D, Guérineau NC, Verleye M, Legros C. (2019). Involvement of the GABAA receptor α subunit in the mode of action of etifoxine. Pharmacol Res. 2019 Jul;145:104250.Epub 2019 May 3.
  7. Hamon A, Morel A, Hue B, Verleye M, Gillardin JM. (2003). The modulatory effects of the anxiolytic etifoxine on GABA(A) receptors are mediated by the beta subunit. Neuropharmacology. 2003 Sep;45(3):293-303.
  8. Shehadeh M, Palzur E, Apel L, Soustiel JF. (2019) Reduction of Traumatic Brain Damage by Tspo Ligand Etifoxine. Int J Mol Sci. 2019 May 29;20(11):2639.

Comment: "Do the authors know whether etifoxine treatment would restore oxidative phosphorylation and mitochondrial function if administered post calcium treatment? The data presented in the manuscript report only on mitochondria pre-treated with etifoxine".

Answer: This comment is particularly pertinent. The reason for the adopted design was time as the longer was the experiment, the weaker became mitochondrial respiration. Since we wanted both exposure to etifoxine and calcium sequentially timed, we thought a pre-treatment model more suitable for the in-vitro experiment, although the cognitive study was based on a post-traumatic treatment paradigm.

Comment: "Several publications have shown that TSPO ligands and TSPO deficiency can inhibit microglial activation and thus influence neuroinflammation. Do the authors know whether the observed effects were seen across all cell types or specific to neurons or microglia".

Answer: This very accurate comment also represents a very logical and plausible explanation for the neuroprotective effect of TSPO, especially when considering the almost absence of neuronal expression of TSPO. Yet, we challenged this hypothesis in previous recent study (Palzur et al., Investigation of the mechanisms of neuroprotection mediated by Ro5-4864 in brain injury. Neuroscience, 2016; 329:162-170. doi: 10.1016/j) and were not able to relate the neuroprotective effect of Ro5-4864 to a direct effect on neuro-inflammation.

Reviewer 2 Report

The manuscript of “Etifoxine restores mitochondrial oxidative phosphorylation and improves cognitive recovery following traumatic brain injury” by E. Palzur et al. is devoted to study a possible neuroprotective effect of etifoxine in the rat model of traumatic brain injury (TBI). The authors found that treatment of rats with etifoxine (50 mg/kg) resulted in a significant improvement of cognitive functions expressed by faster recovery in Morris water maze testing. Based on in vitro experiments, the authors concluded that the neuroprotective effect of etifoxine in TBI may be associated with the and restoration of oxidative phosphorylation capacity due to inhibition of the Ca(2+)-dependent mPTP opening in mitochondria. Unfortunately, the experimental data provided in the work are insufficient to confirm the conclusions drawn in the manuscript. The work requires a series of additional experiments and the addition of control groups in order to be logically completed.

Major comments:

  1. Fig. 1 shows a typical recordings of mitochondrial oxygen consumption in the control group “throughout the successive substrates' additions”. Strictly speaking, only succinate and pyruvate are substrates of mitochondrial respiration. Other supplements are drugs with different mechanisms of action and are not used in classical polarographic studies in the sequence used. The authors should explain why classical uncouplers were not used to study the rate of uncoupling respiration? In addition, the figure does not show the additions of calcium ions and the etifoxine, which are key points for this study. The authors should present typical experiments according to the experimental design described in the Materials and Methods section. It remains unclear why etifoxine was added to the cuvette, and its effects in vivo have not been investigated. What experiments (in addition to behavioral tests) were carried out using post-TBI rats treated with etifoxine?
  2. Fig. 2 should be supplemented with typical curves. Typical curves should be presented for all experimental groups.
  3. The conclusions of the work must be supported experimentally. It is necessary to investigate the in vivo effects of etifoxine on mitochondrial respiration by measuring classical indicators of the efficiency of oxidative phosphorylation (RCR, P/O, time of phosphorylation, etc.) in mitochondria isolated from the brain of treated animals.
  4. The main indicator reflecting the resistance of mitochondria to the mPTP opening is calcium retention capacity (the CRC indicator) (and not the rate of oxygen consumption by mitochondria in the presence of calcium ions). The CRC indicator needs to be measured in mitochondria isolated from the brain of treated animals. If etifoxine does work as an inhibitor of the mPTP, then this indicator should decrease both in the control and experimental groups treated with this drug. Control groups of animals non-treated and treated with etifoxine at the studied (rather high) dose of 50 mg/kg should be included in this study, since the effect of etifoxine as an effective inhibitor of the mPTP opening has not yet been proven. Some recent studies indicate that the peripheral benzodiazepine receptor (Translocator Protein of 18 kDa (TSPO)) is not involved in the regulation of the mitochondrial permeability transition pore opening (DOI: https: //doi.org/10.1074/jbc.M114.549634, etc.).
  5. There are a number of selective blockers of the mPTP opening, which act at nanomolar concentrations and may also serve as therapeutic agents for brain injury. The authors should describe the benefits of using the etifoxine in more detail in the Discussion section.
  6. Lines 321-326. The method used in this work for the isolation of mitochondria seems to be non-standard, since low centrifugation speeds (6,200g for 10 min) and rather small volumes of isolation and washing media (2 ml) were used. The authors need to clarify which structure of the rat brain was used for analysis, since it is difficult to imagine that a homogenate from a whole rat brain weighing about 1.5 g (if the weight of the animals was about 300-350 g) could fit in 2 ml Eppendorf tube. The authors should indicate the concentration of mitochondrial protein in the final suspension of mitochondria, and explain why only a small part of the mitochondrial suspension (0.15 mg/ml) was taken for polarographic studies (0.15 mg/ml as indicated on Line 326, or 0.25 mg/ml as indicated on Line 336?). Was the rest of the mitochondrial suspension used?
  7. Whether the drug can be considered as a neuroprotective agent if its effect on cognitive outcomes in the Morris water maze completely disappears within a short period of time (three days, Fig. 3)?

Minor comments:

  1. Line 17. Abstract. The abbreviation CCI is not decrypted.
  2. Lines 59-60. The sentence is logically incorrect. The sentence begins with words “as a physiological protective mechanism”, but ends with a description of pathological processes in mitochondria, which can lead to the mPTP opening and cell death.
  3. Section 5.4. Mitochondrial respiration measurements. Two different temperatures (30°C and 37°C) are indicated. Please explain.

Author Response

General comment: "The manuscript of “Etifoxine restores mitochondrial oxidative phosphorylation and improves cognitive recovery following traumatic brain injury” by E. Palzur et al. is devoted to study a possible neuroprotective effect of etifoxine in the rat model of traumatic brain injury (TBI). The authors found that treatment of rats with etifoxine (50 mg/kg) resulted in a significant improvement of cognitive functions expressed by faster recovery in Morris water maze testing. Based on in vitro experiments, the authors concluded that the neuroprotective effect of etifoxine in TBI may be associated with the and restoration of oxidative phosphorylation capacity due to inhibition of the Ca(2+)-dependent mPTP opening in mitochondria. Unfortunately, the experimental data provided in the work are insufficient to confirm the conclusions drawn in the manuscript. The work requires a series of additional experiments and the addition of control groups in order to be logically completed."

Answer: In first place, the authors would like to express their gratitude for this in-depth valuable review of our manuscript. Re-reading of the paper under the light of these comments helped us to put things in perspective and concluded that our discussion was reaching too far regarding the possible impact of etifoxine on mPTP opening especially in view of the evidence provided. In fact, our purpose was more to show that the neuroprotective effect clearly established based on the evidence provided in the present as well as in our previous study could be related to improved mitochondrial function, while suggesting a plausible explanation for the mitochondrial protective effect. Consequently, and regardless the new supportive additional data, we rephrased the discussion in order to limit the conclusions to the more established facts.  

Major comments:

Comment: "Fig. 1 shows a typical recordings of mitochondrial oxygen consumption in the control group “throughout the successive substrates' additions”. Strictly speaking, only succinate and pyruvate are substrates of mitochondrial respiration. Other supplements are drugs with different mechanisms of action and are not used in classical polarographic studies in the sequence used. The authors should explain why classical uncouplers were not used to study the rate of uncoupling respiration?

Answer: The protocol used was a short protocol for determining O2 flux in isolated mitochondria. In this protocol, succinate is added without rotenone to measure the mitochondrial respiration's LEAK state (state 4 respiration compensating for the proton leak). After that, ADP is titrated until saturation, followed by titration of pyruvate as a substrate and rotenone to inhibit the complex 1 (NADH oxidation) to determine the mitochondria's respiratory capacity in the ADP-activated state of oxidative phosphorylation. Finally, the addition of antimycin A (inhibitor of CIII) enables us to subtract the residual oxygen consumption due to oxidative side reactions from the oxygen flux as a baseline for all respiratory states to obtain the actual mitochondrial respiration.

Comment: In addition, the figure does not show the additions of calcium ions and the etifoxine, which are key points for this study. The authors should present typical experiments according to the experimental design described in the Materials and Methods section. It remains unclear why etifoxine was added to the cuvette, and its effects in vivo have not been investigated. What experiments (in addition to behavioral tests) were carried out using post-TBI rats treated with etifoxine?"

Answer: Regarding the graph, calcium and etifoxine additions are not mentioned in the typical graph as it represents a control assessment where neither calcium nor etifoxine were added to the assay buffer. In the additional graphs added (see below), calcium and etifoxine additions are now indicated. These additions were in fact performed before suspension of isolated RBM in the assay buffer at T0 time. Also the typical graph has been relocated as suggested.

As for the final sub-comment addressing the in-vitro section of our study, although perfectly true, this comment does not take into account, however, that an entire study was performed and actually previously published in the IJMS exploring and demonstrating the in-vivo effect of etifoxine in traumatic brain injury.      

Comment: "Fig. 2 should be supplemented with typical curves. Typical curves should be presented for all experimental groups"

Answer: We cannot agree more with this comment as it was our first impression though we later deleted these graphs as some of us though that it would unnecessarily inflate paper materials. The graphs have now been reinserted to the revised manuscript as suggested

Comment: "The conclusions of the work must be supported experimentally. It is necessary to investigate the in vivo effects of etifoxine on mitochondrial respiration by measuring classical indicators of the efficiency of oxidative phosphorylation (RCR, P/O, time of phosphorylation, etc.) in mitochondria isolated from the brain of treated animals"

Answer: Although there is little argument with the pertinence of this comment, we thought that the in-vitro model would be more physiologically demonstrative. We think to further perform in-vivo experiments in the future, although it is very clear that this could not be performed in a timely fashion within a 10 days window. Nevertheless, we would like to referrer this reviewer to our previous study that did assess some important mitochondrial functional aspects by ex-vivo experiments such as mitochondrial transmembrane potential, that was found to be partially preserved from the consequence of traumatic brain injury in etifoxine-treated rats.

Comment: "The main indicator reflecting the resistance of mitochondria to the mPTP opening is calcium retention capacity (the CRC indicator) (and not the rate of oxygen consumption by mitochondria in the presence of calcium ions). The CRC indicator needs to be measured in mitochondria isolated from the brain of treated animals. If etifoxine does work as an inhibitor of the mPTP, then this indicator should decrease both in the control and experimental groups treated with this drug."

Answer: The main purpose of our study was to demonstrate that a presumed mitochondrial protective effect induced by etifoxine (expressed by improved oxidative phosphorylation) could promote brain tissue survival and functional recovery from injury. Although we did assume that this effect was allowed by preservation of mitochondrial membrane integrity, the most important observation was the preservation of oxidative phosphorylation as it is the sine qua non condition for ATP production, essential for ionic and water homeostasis. The comment is nonetheless right to the point since assessment of the functional status of respirating mitochondria could only provide an indirect observation. Therefore, in order to both address this comment and comply with the 10 days allowed for submission of the revised manuscript, we added to the study data an analysis of calcium-induced mitochondrial swelling according the technique initially described by Halestrap and Quinlan (Quinlan et al., 1983) and in accordance with the model described in the cited reference within the comment. In this experiment, mitochondrial volume changes were expressed and quantified by decrease in light scattered at 540 nm with 0.5 mg/ml mitochondria suspended in assay buffer supplemented with 2.5 mM succinate and 1 μM rotenone following addition of calcium at 50, 150 and 300 µM, one at a time. Calcium retention capacity could be assessed as the time elapsed from calcium addition (T=0) to maximum decrease in scattered light. As shown in the newly added figure 5, addition of etifoxine to the assay buffer resulted in significant increase in calcium retention capacity. Although calcium concentrations in the buffer assay were not performed to evidence the rupture of mitochondrial membrane, the data is nevertheless supportive of etifoxine impact on mitochondrial membrane permeabilization.

Comment: "Control groups of animals non-treated and treated with etifoxine at the studied (rather high) dose of 50 mg/kg should be included in this study, since the effect of etifoxine as an effective inhibitor of the mPTP opening has not yet been proven. Some recent studies indicate that the peripheral benzodiazepine receptor (Translocator Protein of 18 kDa (TSPO)) is not involved in the regulation of the mitochondrial permeability transition pore opening (DOI: https: //doi.org/10.1074/jbc.M114.549634, etc.)".

Answer: First, the authors express their gratitude for bringing to their attention that important study suggesting an alternative path of action of TSPO ligands that may interfere with ATP production and explain the results of our past and present studies with the need of mPTP involvement. The discussion was modified accordingly. Yet, the quoted study is in contrast with the results of other studies showing the contrary so that we believe that a more balanced discussion should consider both hypotheses and modified the discussion accordingly.

Comment: "There are a number of selective blockers of the mPTP opening, which act at nanomolar concentrations and may also serve as therapeutic agents for brain injury. The authors should describe the benefits of using the etifoxine in more detail in the Discussion section".

Answer: The only drugs that have been used for neuroprotective agent while being described as potential or established mPTP regulators are cyclosporine A and Ro5-4864. As expelained in the manuscript both drugs were discarded either because of side effects (epileptogenic effect of Ro5-4864) or because of failure to prove clinical benefit (CsA).

Comment: "Lines 321-326. The method used in this work for the isolation of mitochondria seems to be non-standard, since low centrifugation speeds (6,200g for 10 min) and rather small volumes of isolation and washing media (2 ml) were used. The authors need to clarify which structure of the rat brain was used for analysis, since it is difficult to imagine that a homogenate from a whole rat brain weighing about 1.5 g (if the weight of the animals was about 300-350 g) could fit in 2 ml Eppendorf tube. The authors should indicate the concentration of mitochondrial protein in the final suspension of mitochondria, and explain why only a small part of the mitochondrial suspension (0.15 mg/ml) was taken for polarographic studies (0.15 mg/ml as indicated on Line 326, or 0.25 mg/ml as indicated on Line 336?). Was the rest of the mitochondrial suspension used?"

Answer: Isolation of mitochondria with well-preserved function is a prerequisite for the measurement of mitochondrial respiration. In the literature, a wide diversity of buffers with different compositions for sample homogenization can be found and various settings related to centrifugation force and time for the separation of mitochondria. The method we selected took into account the need for functionality rather than purity of the isolated mitochondria. Since high centrifugation speeds may produce a more pure mitochondria sample, we were concerned that we might miss the swelled mitochondria prevalent in traumatic brain injury. Therefore we used the method described by Sumbalova et al., 9.

Regarding the exact concentration of mitochondrial protein in the final suspension of mitochondria, this section of the manuscript was admittedly confusion and revised accordingly, indicating clearly the exact concentration that was 0.25 mg/ml. The rest of the mitochondrial suspension was discarded.

Comment: "Whether the drug can be considered as a neuroprotective agent if its effect on cognitive outcomes in the Morris water maze completely disappears within a short period of time (three days, Fig. 3)?"

Answer: Unlike our previous study, animals were submitted to a mild to moderate injury that was no associated at all with mortality not did produce significant tissue loss, Accordingly, not the final outcome but rather the trend and pace of recovery was the contemplated outcome in this study that did show a significant difference between treated and not treated groups.

 Minor comments:

  1. Line 17. Abstract. The abbreviation CCI is not decrypted: abbreviation decrypted.
  2. Lines 59-60. The sentence is logically incorrect. The sentence begins with words “as a physiological protective mechanism”, but ends with a description of pathological processes in mitochondria, which can lead to the mPTP opening and cell death: Indeed, calcium chelation by the mitochondria is a physiological protective mechanism that eventually may lead to mitochondrial damage as calcium buffering capacity of the mitochondria is overwhelmed..
  3. Section 5.4. Mitochondrial respiration measurements. Two different temperatures (30°C and 37°C) are indicated. Please explain: manuscript corrected.

Round 2

Reviewer 2 Report

  1. Authors did not conduct additional experiments
  2. An in vitro model could not be more physiologically indicative. It is not clear on what basis the authors draw a conclusion about the mitochondrial mechanism of the therapeutic action of etifoxine in traumatic brain injury.
  3. It is necessary to investigate how etifoxine affects mitochondrial function in traumatic brain injury.
  4. A number of experimental conditions are not typical for the study of mitochondria in vitro. For example, the concentration of added Ca2+ or etifoxine (80 mM can significantly increase the osmolality of the medium). What solvent was used for etifoxine? It is also not clear at what wavelength the authors measured mitochondrial swelling.

Author Response

Answers to Reviewer 2 (round 2)

Comment: "Authors did not conduct additional experiments"

Answer: This comment is clearly inaccurate as we did add to the revised version of our manuscript a calcium-induced mitochondrial swelling assay, considered by its authors (Quinlan) as a surrogate assessment of mPTP induction. Nevertheless, we would like to attach to this answer the results of a previously conducted experiment which we did not think to add as it appeared to be somehow redundant. However, considering the recurring request for additional data supporting the mitochondrial protective effect of etifoxine, we present this data as attached supplementary material for the purpose of the review. This data is not currently included in the manuscript although we would be willing to add it should this reviewer request it.

The data consists in an ex-vivo analysis of transmembrane mitochondrial potential (Δψm) in RBM isolated from rats submitted to a model to cerebral contusion. Δψm was measured in RBM isolated from tissue specimen obtained from the injured and non-injured hemisphere in control and etifoxine-treated rats, following the same treatment protocol used in the submitted manuscript. As anticipated Δψm levels were similar in grossly intact in both groups in RBM harvested from non-injured tissue whereas there was a significant decrease in Δψm observed in RBM isolated injured tissue of in vehicle-treated animals in comparison with that of etifoxine-treated rats.

Comment: "An in vitro model could not be more physiologically indicative. It is not clear on what basis the authors draw a conclusion about the mitochondrial mechanism of the therapeutic action of etifoxine in traumatic brain injury."

Answer: It seems only fair to say on the view of 3 distinct mitochondrial experiments all showing a clear protective effect including improvement of oxidative phosphorylation, maintenance of Δψm and delayed mitochondrial membrane disruption following exposure to calcium that etifoxine provides a mitochondrial protective effect leading to reduced energy crisis that represent a pivotal event in the generation of secondary post-traumatic cell death.

Comment: "It is necessary to investigate how etifoxine affects mitochondrial function in traumatic brain injury."

Answer: As mentioned above, we have now provided substantial evidence supporting the mitochondrial protective effect of etifoxine. As agreed before, we admittedly omit a calcium retention capacity study usually regarded as a gold standard for mPTP activity analysis and therefore revised our discussion in a more balanced fashion discussing alternative routes of action supported by the same literature quoted by this reviewer.

Comment: "A number of experimental conditions are not typical for the study of mitochondria in vitro. For example, the concentration of added Ca2+ or etifoxine (80 mM can significantly increase the osmolality of the medium). What solvent was used for etifoxine? It is also not clear at what wavelength the authors measured mitochondrial swelling."

Answer: Of course, we did not use a 80 mM concentration of etifoxine which would admittedly have significant consequences on the entire experiment but 80 µM. Unfortunately, the technical notes and protocols of the original manuscript were initially handled by our lab secretary who wrongfully translate "uM" commonly used for µM in with some programs missing special characters, with "mM".

We deeply apologize for our disrespectful cognitive blindness when reading over the manuscript several times. As evidence of our candid mistake, you may as well go through our previous paper (Soustiel, J.F., Zaaroor, M., Vlodavsky, E., Veenman, L., Weizman, A. and Gavish, M. (2008). Neuroprotective effect of Ro5-4864 following brain injury. Exp Neurol 214, 201-208.) using similar experiments with TSPO ligands within the micromolar (Ro5-4864 was used at 100 µM concentration) and not the millimolar range, as it has been our common practice in the lab for years.
